# Adsorption Media for the Removal of Soluble Phosphorus from Subsurface Drainage Water

**DOI:** 10.3390/ijerph17207693

**Published:** 2020-10-21

**Authors:** Jessica K. Hauda, Steven I. Safferman, Ehsan Ghane

**Affiliations:** Department of Biosystems and Agricultural Engineering, Michigan State University, East Lansing, MI 48824, USA; haudajes@msu.edu (J.K.H.); ghane@msu.edu (E.G.)

**Keywords:** agriculture, dissolved phosphorus, eutrophication, nano-engineered media, nonpoint-source pollution, orthophosphate, steel furnace slag, waste materials

## Abstract

Phosphorus (P) is a valuable, nonrenewable resource in agriculture promoting crop growth. P losses through surface runoff and subsurface drainage discharge beneath the root zone is a loss of investment. P entering surface water contributes to eutrophication of freshwater environments, impacting tourism, human health, environmental safety, and property values. Soluble P (SP) from subsurface drainage is nearly all bioavailable and is a significant contributor to freshwater eutrophication. The research objective was to select phosphorus sorbing media (PSM) best suited for removing SP from subsurface drainage discharge. From the preliminary research and literature, PSM with this potential were steel furnace slag (SFS) and a nano-engineered media (NEM). The PSM were evaluated using typical subsurface drainage P concentrations in column experiments, then with an economic analysis for a study site in Michigan. Both the SFS and generalized NEM (GNEM) removed soluble reactive phosphorus from 0.50 to below 0.05 mg/L in laboratory column experiments. The most cost-effective option from the study site was the use of the SFS, then disposing it each year, costing $906/hectare/year for the case study. GNEM that was regenerated onsite had a very similar cost. The most expensive option was the use of GNEM to remove P, including regeneration at the manufacturer, costing $1641/hectare/year. This study suggests that both SFS and NEM are both suited for treating drainage discharge. The use of SFS was more economical for the study site, but each site needs to be individually considered.

## 1. Introduction

Phosphorus (P) is a nonrenewable resource required for all plant growth [1,2]. About 90% of P is used in the global food chain, mainly as fertilizer, and it is estimated that P reserves will be depleted in 50 to 100 years at its current consumption rate [3,4,5]. Total P (TP) can be classified as particulate P (PP) or total soluble P (TSP). TSP remains in the solution after water is filtered using a 0.45-µm filter to remove PP. PP includes living and dead plankton, P precipitates, and P adsorbed to particulate matter [6]. Runoff containing PP often enters surface water, where it settles to the bottom of lakes and streams, making the P less available to algae [7]. P can be further classified as inorganic or organic. Inorganic P is not bound to carbon and hydrogen and includes orthophosphates and polyphosphates. Orthophosphate, also known as soluble reactive P (SRP), is the form used by plants. Polyphosphates are strong complexing agents for metal ions commonly found in detergents and can convert into orthophosphate [6,8]. Organic P is bonded to plant or animal tissues and can be found in excreta and pesticides [6,8].

P losses from agricultural land are a loss of a valuable nutrient. Manure and fertilizers containing P are the main contributors to nonpoint source pollution, both in PP and SRP forms [9]. Other sources include point sources, such as effluent from wastewater treatment plants or industrial facilities [9]. In surface water, this P contributes to eutrophication, including the growth of harmful algal blooms, a higher frequency of hypoxia events, poisonous seafood, losses to aquaculture enterprises, long-term ecosystem changes, and loss of biodiversity [10,11], impacting tourism, human health, environmental safety, and property values [4]. In one case, agricultural runoff partially caused the eutrophication of Lake Erie, leaving upwards of a $100 million annual impact on Ohio’s economy [12,13,14].

Much research has been conducted on PP but only limited on SRP into freshwater bodies. SRP is 95% bioavailable to algae, meaning that it is easily utilized for plant growth, increasing the risk for eutrophication [7]. SRP originates from subsurface drainage or surface runoff [15]. Approximately 18 to 28 million hectares (180,000 to 280,000 km^2^) of cropland in the Midwest region of the USA use subsurface (tile) drainage [16]: perforated drain tubes placed two to four feet (61 to 122 cm) below the crop land that allow water to enter the drain [17]. There is a positive correlation between subsurface drain outlets connected to surface water and the amount of P present in those water bodies [16,18]. P leaching through soil into subsurface drains depends on the drainage system design, soil properties, hydrology, precipitation, season, and land-management practices, such as tillage [16,19].

Subsurface drainage properties can vary from field to field due to different soil types, land management practices, geology, hydrology, and climate [20]. In this research, P adsorption media (PSM) was proposed to capture SRP from agricultural subsurface drainage. Adsorption is the transfer of solutes in a liquid phase, adsorbates, onto a solid adsorbent material, also known as media [21]. Phosphate (PO_4_^3−^) is absorbed to positively charged ions (cations) such as iron, magnesium, calcium, and aluminum by Van der Waal interactions [22].

Examples of PSM include those derived from natural and waste materials and manufactured nano-engineered media (NEM). Different types of PSM exhibit different performance kinetics and adsorption capacities. Natural material-based PSM include zeolite, limestone, and soils. Waste material-based PSM are by-products from other processes such as slags from metal processing plants, fly ash, and water treatment residuals. NEM are manufactured specifically to adsorb phosphorous and have a large surface area and high concentrations of positively charged ions. The base material is typically a ceramic, resin, or biochar that is then coated with metal oxide nanoparticles. These modifications increase the number of adsorption sites on the media, enhancing the PSMs’ overall adsorption capacity [23,24,25,26,27,28]. Table 1 shows different PSM types.

To determine the media that has the best potential to remove SRP from subsurface drainage, the literature was consulted and preliminary studies conducted [39]. These studies entailed testing P removal in subsurface drainage using eight types of PSM: PO_4_Sponge generation 1 and 2, FerrIXA33E, HIX(Zr)-Nano, steel furnace slag (SFS), blast furnace slag (BFS), calcium magnesium biochar, and ferrous sulfate biochar [39]. The PO_4_-Sponge generation 1 and 2, FerrIXA33E, and HIX(Zr)-Nano were types of NEM that were able to remove SRP below detection limits in subsurface drainage with an initial SRP concentration of 0.20 mg/L [39]. The authors also found that the tested NEM had increased removal with increasing media amounts [39]. The BFS and calcium magnesium biochar PSM did not remove SRP from the subsurface drainage [39]. A quantity of 0.15 g of ferrous sulfate biochar removed 21%, 23%, 26%, and 28% after 2, 4, 6, and 24 h of contact time, respectively, at an initial SRP concentration of 0.500 mg/L [39]. The SFS was able to achieve 7%, 12%, and 28% removal using 0.3, 0.6, and 1 g(s) of media, respectively, at an initial SRP concentration of 0.200 mg/L, showing that it was also capable of increased removal with increasing media amounts [39]. Based on this preliminary testing, SFS and PO_4_Sponge, as a representative NEM, which will be referred to as a generalized NEM (GNEM), were selected for further study because of their capacity for a given contact time, hydraulic conductivity, durability, cost, and commercially availability [39].

SFS is a by-product of the steel industry. The slag is formed after lime is injected during the smelting process as a fluxing agent, where it chemically bonds the silicates, aluminum oxides, magnesium oxides, manganese oxides, and ferrites [40]. SFS is then poured, cooled, and processed to remove free metallics and sized for commercial use [40]. The removal mechanism for SFS is a reaction of calcium minerals on the SFS surface with phosphate or bicarbonate ions to produce either calcium phosphate or calcium carbonate, respectively [41]. The optimal conditions for calcium phosphate precipitation are when the pH is 8 or above and there are high concentrations of soluble calcium ions [41,42]. Penn et al. (2020) recommends utilizing SFS to treat subsurface drainage for not more than four to six months due to an accumulation of calcium carbonate minerals on the SFS surface. Calcium carbonates precipitate on the slag instead of calcium phosphate when bicarbonate and dissolved forms of CO_2_ are present in the subsurface drainage, which results from water infiltrating through calcareous soils and microbial respiration [41,42]. This decrease in capacity is due to (1) the bicarbonate and phosphate ions competing to adsorb to the calcium minerals and (2) the decrease in pH due to the formation of calcium carbonate and soluble calcium concentration, which negatively impacts the SFS’s ability to precipitate phosphate ions as calcium phosphate [41,42]. Specifically, the SFS has a decrease in P removal via calcium phosphate precipitation when the pH of the solution is below 8.5 [41,42]. Gonzalez et al. (2020) found that surface runoff was not an issue for the SFS, because it did not contain bicarbonate concentrations causing calcium carbonate mineralization [41].

The 2020 cost of SFS is $0.03/kg [43]. Steel slag fines with a particle diameter of 0.075 mm have a hydraulic conductivity of 6.12 × 10^−3^ cm/s [44]. The specific gravity of SFS ranges from 3.2 to 3.6 [45]. Blanco et al. (2016) used SFS in a batch adsorption study with an initial P concentration of 5-mg P/L and achieved an adsorption capacity of 0.12 to 1.20-mg P/g media [46]. Sheng-gao et al. (2008) conducted a batch adsorption study with SFS using an initial concentration of 1000-mg P/L and achieved an adsorption capacity of 33.3-mg P/g media [47]. Both Sheng-gao et al. (2008) and Blanco et al. (2016) noted that increasing initial P concentrations increased the adsorption capacity of the media [46,47].

The GNEM is a proprietary NEM composed of iron oxide nanocrystals of oxyhydroxide with an alumino-silicate-bonded porous structure containing 80% interconnected pores, a hydraulic conductivity between 3 to 7 cm/s, a base surface area of 15 m^2^/g, a surface area over 70 m^2^/g with nano-modification, and a density of approximately 0.53 g/cm^3^ [22,26,48,49]. Important to note is that the granular form of the GNEM used in this research is only produced for small-scale or laboratory purposes [50]. The manufacturer recommends a monolith of GNEM shaped into cubes, discs, or other shapes for full-scale processes [51]. The GNEM has an adsorption capacity ranging from 25-mg P/g media for low P concentrations (<2 mg/L) and 80-mg P/g media for high P concentrations (>5 mg/L) [30,51]. However, the capacity for very low P concentrations (<0.5 mg/L) is about 10 mg/L [30]. Competing ions found in subsurface drainage are not believed to be a concern for the GNEM [22,50]. P removal is achieved at concentrations from 0.1 mg/L seen in agricultural drainage to 150 mg/L seen in industrial wastewater at food-processing plants [48]. The GNEM can remove P down to levels below 0.09 mg/L for lakes, streams, and agricultural water runoff [30]. Safferman et al. (2015) tested the media using effluent from multiple wastewater treatment plants and found that it reduced SP levels from 1 mg/L to less than 0.3 mg/L [49].

The 2020 cost of the GNEM is $19.00/kg [26]. According to the manufacturer, the GNEM can be regenerated 15 to 20 times, and the regeneration process lowers the average media cost by 80% when compared to the cost of a single, nonregenerated use of this media [48]. Regeneration is the process of stripping ions off the adsorption media using a pH of 10 or higher [21]. P is also easily recovered from the caustic solution as a calcium phosphate precipitate after the regeneration process. For example, Sengupta and Pandit (2011) used sodium chloride and sodium hydroxide to remove phosphate ions off a hydrated ferric oxide (HFO) adsorption media, then calcium or magnesium salt to precipitate phosphate out as a solid-phase fertilizer by-product [52].

Based on the literature review, there is a lack of data and economic considerations on the use of PSM to remove P from subsurface drainage. The objective of this research is to determine the PSM option best suited for managing and removing SRP from a subsurface drainage and gain an understanding of economic factors. This was achieved through the following tasks: (1) characterizing real subsurface drainage water (RSDW) to enable the formulation of synthetic subsurface drainage water (SSDW) for laboratory use studies, (2) running column experiments on selected media to estimate the SRP removal and media capacity under different conditions, and (3) conducting an economic analysis for a study site as a design example and to determine the important implementation consideration.

## 2. Materials and Methods

### 2.1. Study Site Description

The study site was a privately owned farm field in Southeast Michigan, USA that uses subsurface drainage that was installed in 2004 or 2005. Samples for ion analyses were collected from a subsurface drain outlet that drains 6 hectares (14.9 acres). A corn and soybean crop rotation was planted with winter wheat as the cover crop in the winter. Single-rate dry fertilizer was applied in the spring 2019. Data on the RSDW was collected from the study site from January 2019 to August 2019.

### 2.2. Formulating Synthetic Subsurface Drainage

Three samples of RSDW collected from the study site were sent to a commercial laboratory, Merit Laboratories, Inc., East Lansing, MI, USA to perform analyses to determine significant ions related to the use of PSMs. Table 2 contains the average and individual ion concentrations, which were specified for measurement based on the prior literature. The SSDW formulation is in Table 3, which is based on Table 2. The desired P concentration range for the SSDW was determined from this data and literature.

After the SSDW was formulated, two samples were taken to ensure the P concentration was within ±10% of the target concentration.

### 2.3. Column Experiments

Column experiments were conducted to estimate SRP removal and media capacity under different conditions to gain insight on the media performance and data for the study site economic analysis. These column studies can be used for full-scale design, relying on the PSM’s breakthrough capacity [53]. Breakthrough capacity is the amount of adsorbate per mass of adsorbent required to reach the maximum allowable effluent concentration leaving the system, which is typically driven by regulations [53].

A similar concept to breakthrough capacity is saturation. Saturation occurs for an adsorbent when the adsorbate concentration in the effluent is 95% of the adsorbate concentration in the influent, indicating that the adsorbent is filled up with, or saturated with, adsorbate [53]. Breakthrough capacity is not the same as adsorbent saturation. In this research, it was desired to achieve an effluent concentration, breakthrough, that was 50% of the influent.

Two main conditions govern a PSM’s breakthrough capacity: (1) empty bed contact time and (2) influent concentration of the adsorbate. Empty bed contact time (EBCT) is the length of time a volume of solution is in contact with a volume of PSM as it flows through a treatment system [54]. The calculation for EBCT is shown in Equation (1) [53].
(1)$$EBCT=V_{b}/Q$$
where EBCT = empty bed contact time, min, $V_{b}$ = volume of contactor occupied by the PSM, mL, and Q = volumetric flow rate, mL/min.

In addition, the hydraulic retention time (HRT) measures the amount of time required for the solution to flow through a system [55]. Following the method by Hua et al. (2018), the HRT accounts for the porosity of the PSM when it is in a packed bed or column [55]. The porosity of the PSM is important, because the water can only pass through the pore spaces within a packed bed of media (Equation (2)) [56].
(2)$$HRT=\;\frac{V*\xi }{Q}$$
where V = volume of the treatment system, mL, ξ = porosity, and  Q = flow rate going through the column, mL/min.

Figure 1a shows an individual laboratory column, and Figure 1b shows the laboratory columns connected to the influent and effluent tanks. Columns were constructed of PVC pipe with an inner diameter of 3.8 cm (1.5 inches) and a length of 31 cm (12 inches). Each column had a hose barb fitting (1/4″ ID × 1/4″ MIP) in the center of the PVC pipe end cap and another hose barb fitting (1/8″ ID × 1/4″ MIP) roughly 1” from the top of the column. Influent samples were collected from the tubing at the very bottom of the column to eliminate errors associated with concentration changes within the influent storage container and tubing. Effluent samples were collected from the headspace of the column after the solution passed through the bed of media. Influent feed was pumped using Cole-Parmer brand pumps (model no. 7553-70, 7554-80, or 7553-71) from bottom to the top of the column.

The SRP and/or TP concentration in the influent tank and influent testing location were measured to ensure the biofilm did not remove large amounts of P in the tank before entering the column.

To prepare each column, 60 mL of pea gravel (The Home Depot, Store SKU #440773) was rinsed with deionized water (DI) water, dried, and added to the bottom of the column to prevent the media clogging the hose barb supplying the influent. Each media was sieved to a particle size between 1.18 mm and 2.36 mm and rinsed with DI water until the decant water was visually clear before installing into the column. DI water was pumped through the columns with media during flow rate testing to ensure that no ions would adsorb to the media before testing began. During testing, the flow rates for each column were checked daily.

The Hach brand TNT 843 testing method (US EPA equivalent method 365.1 and 365.3) was used to measure SRP after filtering each sample with a 45-µm filter. If the samples could not be tested immediately after collection, they were preserved by adding concentrated sulfuric acid until the pH reached 2 or lower. The samples were stored in a 6 °C (43 °F) covered with plastic film for a maximum of 48 h. To prepare a preserved sample for analyses, the sample was allowed to warm to room temperature (15–25 °C or 59–77 °F); then, the pH of the sample was neutralized using a 5-M sodium hydroxide solution [57]. The volume of both the sulfuric acid to preserve and sodium hydroxide to neutralize were recorded, and the result for that sample was corrected for dilution.

A blank, standard, and replicate at a 10% frequency was included in each sampling event. Data quality was retested or unreported if the percent relative range between replicates was greater than 20% or if the percent recovery for standards was outside a range of 80–110%. For the entire research project, the average percent relative range for replicates and average percent recovery for standards were 2.1% and 101%, respectively, for SRP [39]. These low variations provide confidence that comparisons between column operating conditions could be effectively made, as replicate column studies could not be conducted.

### 2.4. Economic Analysis

An economic analysis was conducted for the study site to demonstrate the protocol and compare the deployment of SFS and GNEM. The study site, described in Section 2.1, served as a case study. Table 4 provides a summary of the flow and SRP concentrations from the study site between October 2018 to July 2019.

The duration of the economic analysis was six consecutive months during which SRP concentrations were at the highest level. This period was also within the recommended range for the use of SFS [48]. The daily flow rate was multiplied by the daily SRP concentration to obtain the daily mass of SRP leaving the subsurface drainage system. The values were summed to obtain the total daily mass of SRP from January 2019 to June 2019, which was 1.66 kg. This is the amount of SRP requiring treatment by the PSM. The average pH between January and June was 7.59 (max = 8.07 and min = 7.20), which is important for the SFS, because calcium carbonate formation starts to occur below a pH of 8 [41].

After the annual treatment period, the GNEM was designed to be regenerated for a total of 15 regenerations (as its estimated lifetime is 15 years). There is no literature that indicates that the SFS can be regenerated, so it is assumed to be annually removed from the treatment site via a vacuum truck and disposal at a recycling facility near the study site. Three scenarios were developed. Figure 2: (1) Scenario A—the cost of the GNEW plus regeneration by the manufacturer, (2) Scenario B—the cost of the GNEW plus onsite regeneration by the farmer on site, and (3) Scenario C—the cost of the SFS plus the cost of disposal.

The cost for each scenario assumed a low inflation and low interest rate. This cost was then divided by the number of acres at the study site to determine the annual cost per hectare/acre. Thereafter, the solver tool in Microsoft Excel was used to determine the break-even cost between the three scenarios by changing the PSM capital cost and the GNEM regeneration cost.

## 3. Results and Discussion

### 3.1. Column Experiments

Column experiments were conducted for the GNEM and SFS at an EBCT of five min. The equivalent HRT for the GNEM and SFS was four and three min, respectively. The initial SRP concentration of the SSDW was 0.50 mg/L. These conditions tested the media capacity and longevity for a worst-case scenario for the case study at the study site by simulating peak flow rates and high initial P concentrations. Table 5 shows the experimentally determined laboratory column characteristics for each PSM column used in this research.

Both the GNEM and SFS removed SRP to below detection limits for up to five hours after the columns were turned on. After 24 h, the SFS was removing approximately 50% SRP, while the GNEM was still removing SRP to below detection limits. After seven days of continuous operation, the SFS was saturated. The GNEM, however, was removing 20% SRP after 10 days of continuous operation. The final saturated adsorption capacity for the GNEM and SFS was 1.176 and 0.162-mg P/g media, respectively. These saturated adsorption capacities were determined by summing the total amount of SRP sorbed and then dividing that value by the total mass of PSM used. Figure 3 and Figure 4 show the influent and effluent concentrations for the GNEM and SFS, respectively. Figure 5 shows the cumulative SRP sorbed throughout the column experiments.

### 3.2. Additional Column Experiments

To determine the mechanism of removal, additional column experiments were run with varying the initial concentration and flow rates, which changed the EBCT (HRT). Table 6 shows the change in EBCT or initial concentration and the type of PSM. Figure 6 and Figure 7 show the influent and effluent SRP concentration changes for the GNEM and SFS, respectively, in response to the changes in EBCT.

For condition “a”, removal was observed for the first three days for both the GNEM and SFS, but no removal was observed after three and five consecutive days for SFS and GNEM, respectively. When the EBCT was doubled from five to 10 min for both types of PSM for condition “b”, no removal was observed for the GNEM or SFS. When the EBCT was doubled from 10 to 20 min for both types of PSM for condition “c”, again, no removal was observed for both types of PSM. When the EBCT was kept at 20 min and the initial SRP concentration was increased from 0.50 to 2.00 mg/L during condition “d”, the GNEM removed about 50% SRP during the first testing period on day 24, but the SRP removal declined over the remaining duration of condition “d” as the PSM became saturated. No SRP removal was observed for the SFS for condition “d”. Note that there were decreasing influent concentrations during condition “d”, which was hypothesized to be caused by the SRP reacting within the feed tank. Lastly, the EBCT was increased from 20 to 60 min for condition “e”. The GNEM and SFS removed 17% and 5%, respectively, during the first testing period; then, both removed 11% during the second testing period. Overall, for conditions “a”–“e”, these trends strongly indicated that both PSM were saturated in condition “a” and that a higher EBCT did not greatly increase the PSM capacity. This finding indicates that the capacity of the media is minimally impacted by the flow rate and the influent SRP typical in subsurface drainage, greatly simplifying site-specific system sizing.

Additionally, the monolith of GNEM was tested, as this configuration is more economical to produce compared to the granular configuration. Research showed that the TP removal from SSDW showed that the monolithic and granular versions of the GNEM had similar performances (Figure 8) when exposed to the same influent concentration [39].

### 3.3. Determination of Media Volume

The target influent SRP concentration for the GNEM and SFS was 0.50 mg/L. A concentration of 0.25 mg/L (50% of the target influent concentration) was selected to represent the breakthrough concentration for the economic analysis. An influent loading method was used to determine the amount of media required for site-specific conditions. The influent loading method entails correlating the effluent concentration to the cumulative mass loading (Figure 9 and Figure 10). The equation for the linear trend line allows for the calculation of the cumulative mass of SRP loaded into the column to reach the target SRP effluent level. This method allows for variable influent flows and concentrations between the laboratory and actual system, especially for the two PSMs selected in this research, as the concentrations typically found in the subsurface drainage did not significantly change the media’s performance (Figure 6 and Figure 7).

Setting the x-value to the breakthrough concentration of 0.25 mg/L results in a y-value from the graph representing the total amount of SRP the PSM adsorbed. Dividing the obtained y-value by the total volume of media used in the column (Table 5) yields the mass of SRP adsorbed per volume of PSM. Dividing the target amount of SRP requiring removal by the mass of SRP per volume of PSM by yields the volume of PSM to remove the target SRP amount. Table 7 contains the required volume of SFS and GNEM to treat 1.66-kg SRP from the study site.

### 3.4. Economic Analysis

For the study site theoretical economic analysis case study, a breakthrough value of 50% of the influent was arbitrary selected to illustrate the protocol. The calculated volume of GNEM and SFS required to remove 1.66-kg SRP from the study site was 4.22 m^3^ and 15.5 m^3^, respectively. Note that the calculated volume of PSM assumed that the entire flow from the study site was able to pass through and be treated by the contactor holding the PSM.

The maximum and minimum EBCT and HRT in Table 8 were calculated from the daily site flow rate data and the required volume of PSM and the media porosity. The porosity of the GNEM was given as 0.80 by the manufacturer [26]. The calculated porosity of the SFS was 0.57 using the laboratory bulk density for SFS from Table 5 and an average specific gravity of 3.4 (assumed specific gravity was the same as particle density based on the literature) [45,56].

Note that the minimum EBCT/HRT is the most critical design consideration, because the PSM must remove SRP from the peak flow rate conditions. In all cases, the values for the EBCT and HRT were longer than the tested values in the laboratory column experiments. The costs associated with implementing the GNEM and SFS at the study site are presented in Table 9. Included in the table is a brief description of each item and the basis for the cost.

Using the costs in Table 9, the annual cost per hectare was calculated for scenarios A, B, and C using the method illustrated in Table 10. The annual cost per hectare calculation was done in an Excel spreadsheet titled “Media_Feasibility_Study_Extension_Manuscript_InfluentLoadingMethod.xlsx”, which can be found in the Appendix A for this article.

The annual cost per hectare to implement the PSM for scenarios “A”, “B”, and “C” were $1641, $1172, and $906, respectively, making scenario “C” the most cost-effective option. Table 11 shows the calculated percent different in the annual cost per hectare and acre for each scenario.

Additionally, Excel’s solver tool was used to determine if changing the GNEM capital or regeneration cost for scenarios “A” and “B” would result in its annual cost per acre to break even with the SFS in scenario “C”. The solver tool determined that the GNEM in scenario “A” breaks even with scenario “C” if both the capital and regeneration cost are $11.34/kg (40% decrease) and $0.73/kg (63% decrease), respectively. The solver tool determined that a GNEM capital cost of $9.32/kg (51% decrease) or a regeneration cost of $0.35/kg (65% decrease) makes scenario “B” break even with scenario “C” for the annual cost per acre. Additionally, the solver tool determined that the GNEM in scenario “B” breaks even with scenario “C” if both the capital and regeneration cost are $13.04/kg (31% decrease) and $0.75/kg (25% decrease), respectively.

In addition to the above analysis, the annual cost per acre for scenarios “A”, “B”, and “C” were compared to the annual revenue per acre of rotational field corn. For a high-productivity soil producing 211 bushels/acre at a harvest price of $3.80/bushel, the expected revenue per acre for rotational corn is $359 [60]. For average and low-productivity soils producing 176 and 141 bushels/acre, respectively, the expected revenue per acre for rotational corn is $246 and $155 per acre, respectively [60]. Based on these expected revenues per acre, none of the scenarios have an associated cost allowing the farmer to realistically implement this technology at the study site without substantial subsidies. However, each site must be individually considered using the protocol and spreadsheet (see Appendix A).

## 4. Conclusions

GNEM removed SRP to very low levels, below the 50-µg/L analytical detection limit, at an EBCT as low as 5 min. The resulting GNEM capacity at these low SRP levels and rapid EBCT was 1.2-mg SRP per g of media. The SFS had a similar performance but only had a capacity of 0.17 mg of SRP per g of media. Other types of media tested did not successfully remove the SRP to the very low target level.

Using the SFS to remove P and then disposing it after use was the most cost-effective option based on a specific study site. Using the GNEM media to remove P and then regenerating it onsite was slightly more expensive than the SFS, with a percent relative difference of 26%. These costs greatly depend on the many site-specific assumptions and are not transferable, but the methodology can be applied to other locations. Additionally, this was a theoretical analysis, as the media was not actually deployed at the study site. However, shipping the GNEM back to a centralized location was comparably very expensive, having a percent relative difference between 33% and 53%, and will likely not be effective for most situations. Further, media costs and the value of the recovered phosphorus is also expected to vary. As GNEM becomes more common, new products will increase competition and manufacturing and performance optimization will likely lower costs. Conversely, as the use of SFS increases, costs may increase. However, if disposal of the SFS on-site is feasible or it can be used as a beneficial soil amendment and SRP recovered, costs for disposal will be eliminated.

Although implementation costs were found to be equal or greater than crop revenues for the study site case study, there are opportunities for efficiencies, and governmental conservation incentives may be possible. Further, a more optimal design approach should be explored. The very simplistic approach used for the study site results in SRP concentrations always below the breakthrough level of 50% of the influent concentration. This results because the equilibrium mechanisms of the PSM result in greater SRP removal when there is less SRP adsorbed to the media and less removal when more is adsorbed. A system where two columns of PSM are used would reduce the overall required amount of media. The first column would be a roughing system, and the second would lower the SRP to detection limits. Once SRP levels start to increase, this second column would be switched to the first position and serve as a roughing system, and new SFS or regenerated GNEM would be used in the polishing (second) column. This SRP-free subsurface drainage water would then be blended with equal amounts of untreated subsurface water, which would theoretically maintain the blended effluent to below the breakthrough target of 50% for the case study. However, this would be at the expense of more complex operations, which may negate any benefits.

## Figures and Tables

**Figure 1 ijerph-17-07693-f001:**
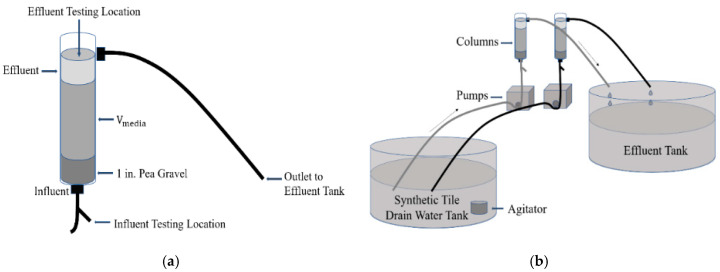
(**a**) Laboratory column and (**b**) diagram of phosphorus sorbing media (PSM) columns connected to the influent and effluent tanks.

**Figure 2 ijerph-17-07693-f002:**
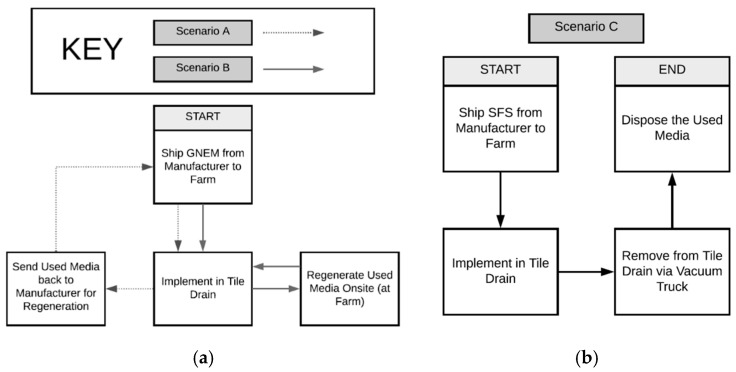
(**a**) Flow diagram for scenarios A and B, and (**b**) flow diagram for scenario C. GNEM: generalized nano-engineered media and SFS: steel furnace slag.

**Figure 3 ijerph-17-07693-f003:**
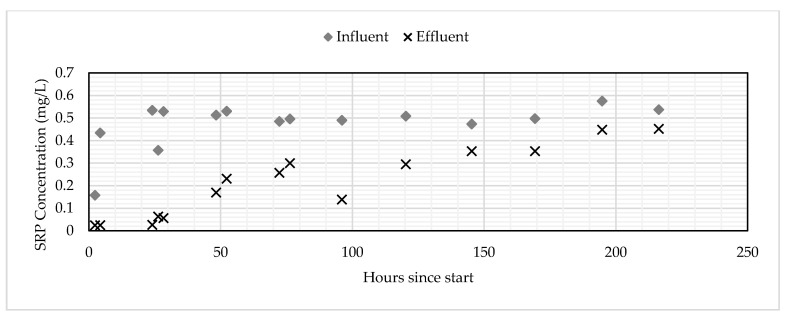
Influent and effluent concentrations for the GNEM. SRP: soluble reactive phosphorous.

**Figure 4 ijerph-17-07693-f004:**
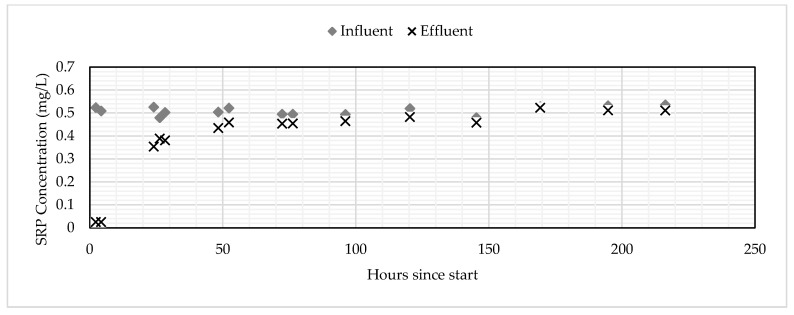
Influent and effluent concentrations for the SFS.

**Figure 5 ijerph-17-07693-f005:**
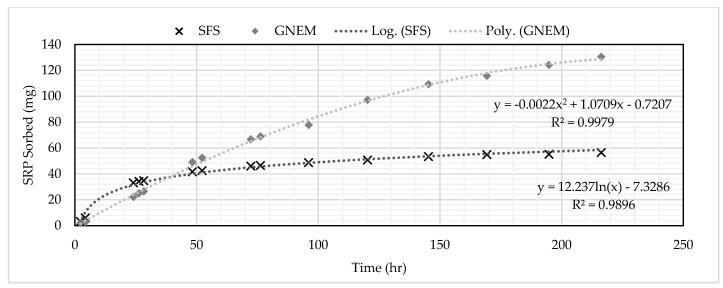
Comparison of SRP loading onto the GNEM and SFS media.

**Figure 6 ijerph-17-07693-f006:**
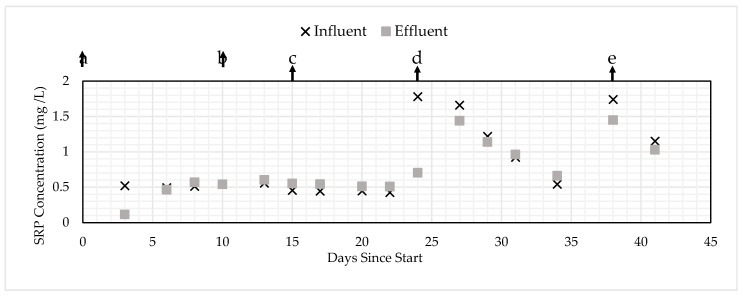
SRP removal for the GNEM under changing initial concentrations and empty bed contact times/hydraulic retention times (EBCTs/HRTs).

**Figure 7 ijerph-17-07693-f007:**
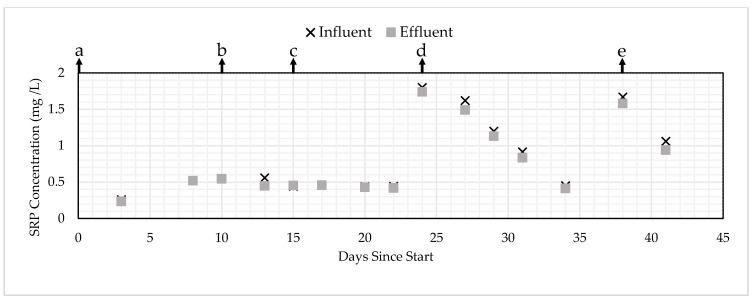
SRP removal for the SFS under changing initial concentrations and EBCTs/HRTs.

**Figure 8 ijerph-17-07693-f008:**
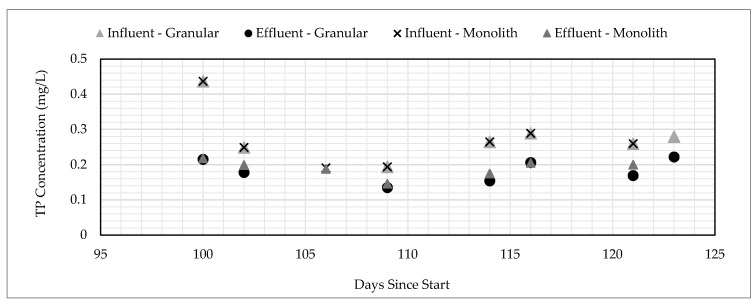
Comparison of phosphorous (P) removal by the monolithic and granular GNEM.

**Figure 9 ijerph-17-07693-f009:**
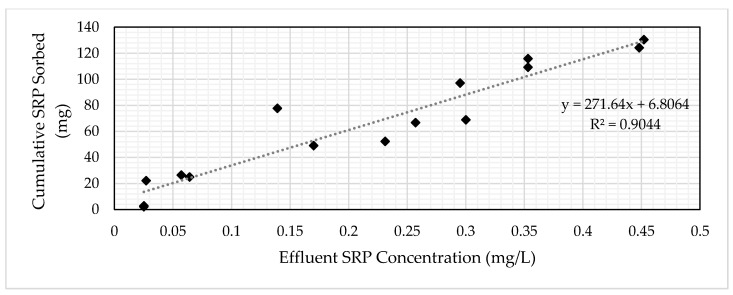
SRP concentration vs. cumulative SRP-sorbed graph for GNEM.

**Figure 10 ijerph-17-07693-f010:**
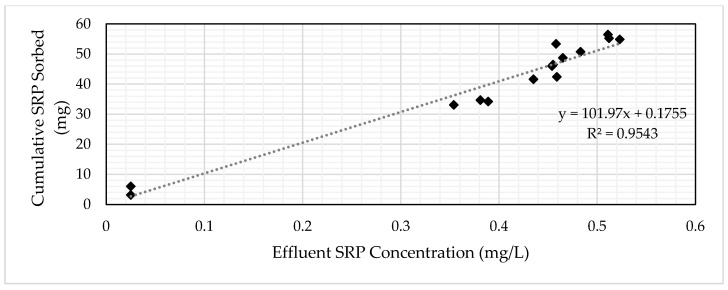
SRP concentration vs. cumulative SRP-sorbed graph for SFS.

**Table 1 ijerph-17-07693-t001:** Different types of natural, waste, and nano-engineered phosphorous (P) adsorption media.

Name	Type	Adsorption Capacity	Initial Concentration	Form	Water Type	Reference
Limestone	N	0.68 mg-P/kg	40 mg/L	PO_4_-P	DI water and H_2_KPO_4_	[29]
PO_4_Sponge	NEM	80,000 mg-P/kg	>5 mg/L	P	Wastewater and agricultural runoff	[30]
		50,000 mg-P/kg	<2 mg/L	P		
Al-treated Steel Slag	NEM	53 mg-P/kg	0.16 to 2.3 mg/L	SP	Subsurface drainage	[31]
FerrIXA33E	NEM	2300 mg-P/kg	0.260 mg/L	P	Wastewater	[32]
Zeolite	N	0.46 mg-P/kg	40 mg/L	PO_4_-P	DI water and H_2_KPO_4_	[29]
Serpentinite	N	1.37 mg-P/kg	20 mg/L	P	DI water and H_2_KPO_4_	[33]
Natural Soils	N	6.3 to 501.0 mg-P/kg	3.3 mg/L	PO_4_-P	DI water and H_2_KPO_4_	[34]
Dolomite	N	52 mg-P/kg	60 mg/L	PO_4_-P	DI water and H_2_KPO_4_	[35]
Banana Straw Biochar	NEM	3115 mg-P/kg	250 mg/L	TP	DI water and H_2_KPO_4_	[36]
Electric Arc Furnace Slag	W	2.51 mg-P/kg	20 mg/L	P	DI water and H_2_KPO_4_	[33]
Fly Ash	W	0.86 mg-P/kg	40 mg/L	PO_4_-P	DI water and H_2_KPO_4_	[29]
Blast Furnace Slag	W	0.006 mg-P/kg	0.180 mg/L	P	DI water and H_2_KPO_4_	[37]
Filtralite P™	NEM	2500 mg-P/kg	0.480 mg/L	PO_4_-P	DI water and H_2_KPO_4_	[38]
D-201	NEM	1220 mg-P/kg	10 mg/L	PO_4_-P	DI water and H_2_KPO_4_	[25]
HFO-201	NEM	17,800 mg-P/kg	10 mg/L	P	DI water and H_2_KPO_4_	[25]

N = natural, W = waste, and NEM = nano-engineered media. DI: deionized.

**Table 2 ijerph-17-07693-t002:** Summary of ion analyses for three real subsurface drainage samples from the study site.

Chemical	Measured Concentration (mg/L)			
	RSDW #1	RSDW #2	RSDW #3	Average
SO_4_^2−^	251	138	123	171
Cl^−^	14	14	15	14.3
NO_3_-N	7.5	7.1	9.8	8.13
SiO_2_ ***	14	14.5	14	14.2
Ca^2+^	206	180	171	186
Mg^2+^	50.9	39.6	40.4	43.6
K^+^	under range *	3.49	2.91	3.2 **
Na^+^	15.2	11.1	13.7	13.3
Avg. Daily Flow (m^3^/d)	6.69	0.13	0.04	2.29

* Under range for potassium was classified as <2.5 mg/L. ** The under-range value for potassium was not accounted for in the average value for potassium. *** Standard Method 4500 Si D 2011; the commercial laboratory also used methods “SW 846 Method 3015A Revision 1 February 2007”, “EPA Method 300.0 Revision 2.1”, and “EPA Method 200.8 Revision 5.4”. RSDW: real subsurface drainage water.

**Table 3 ijerph-17-07693-t003:** Chemical compound concentration in the synthetic subsurface drainage formulation.

Chemical	Target Concentration (mg/L)			
	0.200	0.500	1.00	2.00
KCl	5.62	4.90	3.69	1.29
MgSO_4_	106.93	106.93	106.93	106.93
CaSO_4_	120.93	120.93	120.93	120.93
NaNO_3_	49.35	49.35	49.35	49.35
NaCl	19.22	19.79	20.73	22.62
Si(OH)_4_	14.17	14.17	14.17	14.17
H_2_KPO_4_	0.88	2.20	4.39	8.79

**Table 4 ijerph-17-07693-t004:** Summary of the flow rate and soluble reactive phosphorous (SRP) concentration range in subsurface drainage at the study site between October 2018 and July 2019.

Month	Flow Rate Range (m^3^/day)	SRP Range (mg P/L)	Month	Flow Rate Range (m^3^/day)	SRP Range (mg P/L)
October	0–13.51	0–0.05	March	1.57–865.27	0.004–0.093
November	11.73–936.63	0.003–0.050	April	16.46–1134.34	0.003–0.332
December	4.29–848.73	0.003–0.061	May	19.23–1009.4	0.0001–0.305
January	1.42–551.60	0.003–0.132	June	2.10–1253.40	0–0.070
February	6.64–446.06	0.014–0.481	July	0–39.1	0.004–0.168

**Table 5 ijerph-17-07693-t005:** Measured generalized nano-engineered media (GNEM) and steel furnace slag (SFS) column characteristics. EBCT: empty bed contact time and HRT: hydraulic retention time.

Parameter	GNEM	SFS
Volume (mL)	190	240
Mass (g)	111.7	353.5
Bulk Density (g/cm^3^)	0.588	1.473
EBCT (min)	5	5
HRT (min)	4	3

**Table 6 ijerph-17-07693-t006:** Additional column experimentation and respective conditions.

Condition	Day	Initial P Concentration (mg/L)	GNEM			SFS		
			EBCT (min)	HRT (min)	Removal	EBCT (min)	HRT (min)	Removal
a	0	0.500	5	4	Yes	5	3	Yes
b	10	0.500	10	8	No	10	6	No
c	15	0.500	20	16	No	20	11	No
d	24	2.000	20	16	Yes	20	11	No
e	38	2.000	60	48	Yes	60	34	Yes

**Table 7 ijerph-17-07693-t007:** Measured GNEM and SFS column characteristics. PSM: phosphorus sorbing media.

Parameter	GNEM	SFS
Equation	y = 101.97x + 0.1755	y = 271.64x + 6.8064
Volume (mL); x-value	190	240
Mass of SRP per volume PSM (mg/mL PSM)	0.39	0.11
Total volume of PSM required to remove 1.66-kg SRP (m^3^)	4.22	15.5

**Table 8 ijerph-17-07693-t008:** The minimum and maximum EBCT and HRT for the GNEM and SFS media for the study site.

Month.	GNEM		SFS	
	EBCT (min)	HRT (min)	EBCT (min)	HRT (min)
January	10–4060	8–3250	42–16300	24–9300
February	13–869	10–695	52–3500	30–1990
March	7–3680	5–2950	27–14800	15–8450
April	5–351	4–281	20–1410	12–804
May	6–300	5–240	23–1210	13–689
June	5–2690	4–2150	19–10800	11–6170

**Table 9 ijerph-17-07693-t009:** Costs associated with PSM implementation [39].

Item	GNEM	SFS	Basis
Media Capital Cost	$19/kg	$0.05/kg ^1^	Obtained from manufacturer
Contactor Capital Cost	$6685	$10,869	Includes capital and shipping cost to study site
Labor Cost	$480	$480	Two contractors, $30/h, 8 h/day
Contactor Installation Cost	$640	$640	One backhoe with an operator $80/h, 8 h/day
Shipping from Manufacturer to Study Site	$201	N/A	
Shipping from Study Site to Manufacturer	$169	N/A	
Regeneration Cost by Manufacturer	$2/kg	N/A	Provided by manufacturer
Onsite Regeneration Cost	$1/kg	N/A	Assumed to be half the cost of manufacturer regeneration
Cost of Recovered P	$0.002/g	N/A	[58,59]
Vacuum Truck Rental	N/A	$725/truck/day	Truck Rental Businesses near study site
Cost of Diesel Fuel in Midwest	N/A	$2.978/gallon	https://www.eia.gov/petroleum/gasdiesel/ for June 2019
Vacuum Truck Diesel Fuel Tank Volume	N/A	113 gallons ($337 per tank)	Truck Rental Businesses near study site
Vacuum Truck Mileage	N/A	7.5 MPG	Truck Rental Businesses near study site
Disposal Site Entrance Fee	$12/day	$12/day	The entrance fee cost for the recycling center near the study site
General Waste Disposal Cost	$28/yd^3^	$28/yd^3^	The cost to dispose each cubic yard of material at the recycling center

^1^ Includes the capital and shipping cost for the media.

**Table 10 ijerph-17-07693-t010:** Costs from Table 9 included in the determination of the annual cost per acre to implement the PSM for year 0, 1–14, and 15.

Item/Year	Scenario A			Scenario B			Scenario C		
	0	1–14	15	0	1–14	15	0	1–14	15
Media Capital Cost	×	-	-	×	-	-	×	×	×
Contactor Capital Cost	×	-	-	×	-	-	×	-	-
Contactor Installation Cost	×	-	-	×	-	-	×	-	-
Labor to Install Media in Contactor	×	×	-	×	×	-	×	×	-
Shipping Cost	×	×	×	×	-	×	-	-	-
Regeneration Cost	-	×	×	-	×	×	-	-	-
Labor to Remove Media from Contactor	-	×	×	-	×	×	-	×	×
Removal Cost via Vacuum Truck	-	-	-	-	-	-	×	×	×
Disposal Cost	-	-	-	-	-	-	×	×	×

“×” = included and “-” = not included.

**Table 11 ijerph-17-07693-t011:** Calculated percent difference between the annual cost per hectare and acre for scenarios A, B, and C.

Scenario	A	B	C
Annual Cost Per Hectare ($/hectare/year)	$1641	$1172	$906
Annual Cost Per Acre ($/acre/year)	$664	$474	$367
% Difference Compared to Scenario A	N/A	33%	53%
% Difference Compared to Scenario B	33%	N/A	26%
% Difference Compared to Scenario C	53%	26%	N/A

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
