# Peer review of "Adsorption Media for the Removal of Soluble Phosphorus from Subsurface Drainage Water"

_ijerph, 2020, doi:10.3390/ijerph17207693_

Round 1

Reviewer 1 Report

Line 46 - "poisonous seafood" seems to be not a direct effect of eutrophication, perhaps more of heavy metals content in seawater.

Line 54-55 - hectares, feets etc. I would recommend adding metric system units in brackets

Line 61 and many many more - "Error! Reference source not found" the manuscript needs some work with the used automatic reference software.

Line 84 (Table 3) - NE=Nano-Engineered Media when in the Table you used "NEM"

Line 170 - double space

Line 369 (Table 11)

For Vacuum Truck Diesel Fuel Tank Volume N/A 113 gallons - gallons of what? can you calculate it to $$, same with Vacuum Truck Mileage MPG

what does $28/yd3 means? 

Line 403 - the conclusions are very general and need to include some numbers from the results of SFS and GNEM analysed properties.

Author Response

Reviewer 1

Thank you for the very thoughtful review.  Below are our responses to each comment and the resulting changes that were incorporated into the revised manuscript.

No.

Comment Given

Action Taken/Commentary

1

Line 46 - "poisonous seafood" seems to be not a direct effect of eutrophication, perhaps more of heavy metals content in seawater.

We are just reporting text (“poisonous seafood”) from published literature (#10 in the bibliography). I believe the author meant that seafood could be rendered inedible as an indirect consequence of eutrophication (as you pointed out.

2

Line 54-55 - hectares, feets etc. I would recommend adding metric system units in brackets

Added metric units in parentheses in lines 54 and 55 of revised manuscript.

3

Line 61 and many more - "Error! Reference source not found" the manuscript needs some work with the used automatic reference software.

Thank you for pointing this out.  This was caused by Endnotes.  All are now fixed.

The cross-references for tables and figures were also removed so conversion to reference errors in the future would not occur.

4

Line 84 (Table 3) - NE=Nano-Engineered Media when in the Table you used "NEM"

Thank you for catching this – corrected to “NEM’ on line 76 of the revised manuscript.

5

Line 170 - double space

Corrected, thank you for catching this.

6

Line 369 (Table 11)

For Vacuum Truck Diesel Fuel Tank Volume N/A 113 gallons - gallons of what? can you calculate it to $$, same with Vacuum Truck Mileage MPG

what does $28/yd3 means?

The ‘For’ was not supposed to be there, thank you for catching that (Line 361 in revised manuscript).

The cost associated with each full tank of diesel fuel is in parentheses, Line 359, Table 9 in the revised manuscript.

The $28/yd3 was the disposal fee per cubic yard of waste brought to the recycling center. The description is updated, Line 359, Table 9 in the revised manuscript.

7

Line 403 - the conclusions are very general and need to include some numbers from the results of SFS and GNEM analysed properties.

More details have been added in the conclusions including the amount of SRP that was reduced, the EBCT, and capacity (Lines 393-397 in revised manuscript).  Further, numerical values were added on the cost differences on Lines 400 and 404 of the revised manuscript.  A more numerical statement was added on line 410 of the new manuscript.

Reviewer 2 Report

In the abstract, why the author has mentioned tourism in this context? Is that a typo?

On page 2, line 62: reference error. Also, it is recommended to rephrase the paragraph. A better way of citing others' work is to discuss passively other data and provide a reference number at the end of the discussion.

Page 3, line 73: typo in reference number 33

Page 3: typo in Byproducts, (by-product)

Page 3, line 83: reference error

The introduction is too long! 4 pages of the manuscript are discussing others' work. The introduction needs to be revised.

It is recommended to revise Table 3 since several similar works have been cited. It is suggested that write the manuscript in a more concise format and avoid repeating citations that are not drastically different. For Table 3, one example of each type can deliver the idea.

Page 5, 6, 10 etc : Reference error

The analytical method and its instrument for analyzing vales in Table 4 should be provided.

Typo table 8 “?”

Author Response

Reviewer 2

Thank you for the very thoughtful review.  Below are our responses to each comment and the resulting changes that were incorporated into the revised manuscript.

No.

Comment Given

Action Taken/Commentary

1

In the abstract, why the author has mentioned tourism in this context? Is that a typo?

The Great Lakes are a popular destination for tourism, but eutrophication can deter people from swimming, boating, etc. due to the color of the water, smell, and even toxicity.

2

On page 2, line 62: reference error. Also, it is recommended to rephrase the paragraph. A better way of citing others' work is to discuss passively other data and provide a reference number at the end of the discussion.

Fixed reference error, Line 61 on the revised manuscript.

We apologize as am not sure on the exact recommendation.  Perhaps it concerns Table 1.  If so the authors feel a table of the existing data most clearly and concisely represents the literature.

3

Page 3, line 73: typo in reference number 33

Duplicate reference was removed, Line 64 in revised manuscript.

4

Page 3: typo in Byproducts, (by-product)

All instances of “byproduct” has been changed to “by-product”.

5

Page 3, line 83: reference error

All reference errors have been corrected.

6

The introduction is too long! 4 pages of the manuscript are discussing others' work. The introduction needs to be revised.

Removed tables 1 and 2 from the introduction between lines 61 and 68.  This shortened the Introduction by ~1 page.  Carefully read through the rest of the Introduction and feel the details significantly support the project objectives.

7

It is recommended to revise Table 3 since several similar works have been cited. It is suggested that write the manuscript in a more concise format and avoid repeating citations that are not drastically different. For Table 3, one example of each type can deliver the idea.

Thank you for this comment but Table 3 (now Table 1 in the revised manuscript) is crucial as a project objective was to find the best PAM and a thorough review of the literature was essential and will be important for practitioners.   

8

Page 5, 6, 10 etc : Reference error

All reference errors have been corrected.

9

The analytical method and its instrument for analyzing vales in Table 4 should be provided.

Added the laboratory methods, Lines 165 – 168 in the revised manuscript, and commercial laboratory used, Line 159 in revised manuscript. 

10

Typo table 8 “?”

Removed, Line 288, Table 6 in the revised manuscript.